



# Comparison of different simulation methods regarding loads, considering the Center of Wind Pressure

Marcel Bock[1], Daniela Moreno[1], Jan Friedrich[1], and Joachim Peinke[1]

[1]ForWind – Institute of Physics, University of Oldenburg, Germany

**Correspondence:** Marcel Bock (marcel.bock@uol.de)

**Abstract.** This study presents a comprehensive comparison of different wind turbine simulation methods, with a focus on aerodynamic load prediction using the concept of the Center of Wind Pressure (CoWP) [C. Schubert et al., Wind Energ. Sci. Discuss., 2025]. Simulations under laminar, shear, and turbulent inflow conditions are carried out for this comparison. A new quantity, namely the Load center, is introduced to correlate the flow-related CoWP and the loads of the turbine. A novel calibration factor is introduced to establish a direct relationship between flow structures and aerodynamic loads. A good correlation between the inflow wind field and loads from blade element momentum simulations (BEM) is found. High-resolution Large eddy simulations (LES) show improved correlation with CoWP-based load estimates, attributable to the more-resolved flow modelling capabilities.

## 1 Introduction

Installations of new, state-of-the-art wind turbines have to be carried out in accordance with current standards, e.g., the IEC61400-1 IEC (2019). The standards cover most aspects of turbines over their service life. This includes various operating points such as regular power generation, start-up phase, normal shutdown, and error handling. Some of these design load cases have to be tested under a whole range of wind speeds. In total, several hundred different cases have to be analysed for compliance with the standards. Due to the enormous number of cases, it is necessary to use efficient tools that can deliver accurate results in the shortest possible time.

In general, there are various techniques for simulating wind turbines. The most common are the blade element momentum theory (BEM), actuator line simulations (LES-AL), and blade resolved simulations (LES-BL). These methods differ significantly in terms of complexity and calculation effort. The simplest method is BEM, an engineering model in which the local velocities are estimated from an induction model and the resulting blade forces are calculated using lookup tables. Since simple surrogates of the flow field are used, BEM simulations are computationally efficient. LES-AL and LES-BL are computational fluid dynamics simulations (CFD) in which the flow field around the turbine is calculated by solving the Navier-Stokes equations. Usually, a large eddy simulation (LES) is used for modelling the turbulence. Thus, the impact of the turbulent inflow cases is not just treated by an induction factor. Still, the spatio-temporal development of the turbulent flow structures is resolved as they approach the turbine. On the one hand, LES simulations allow very accurate predictions of the interaction between the blades and the flow. On the other hand, LES simulations are orders of magnitude more costly than BEM. Accordingly, it would be





impossible to simulate several hundred load cases for validation processes as part of the development and optimization of wind turbines with computational capabilities. This is why BEM forms the basis for the development process.

This raises the question: How accurate are the predictions using BEM compared to high-resolution LES? Due to the lack of flow modelling in the induction zone, a wind gust can disappear or be strongly deformed until it hits the rotor. The flow field
immediately after the rotor can also affect the local blade aerodynamics. All of these phenomena can occur in reality and can be modelled with LES, but cannot be represented in BEM. Hence, to evaluate such model uncertainties, comparative studies are of high interest.

The following paragraph summarises existing comparisons from the literature. In Ehrich et al. (2018) the effects of turbulence on the sectional forces are analysed for BEM, LES-AL, and BL. It was concluded, that the sectional forces for the center
section of the blade match between the simulation methods, but differ at the blade root and tip. Liu et al. (2022) compared the power and thrust in BEM and LES-AL for laminar inflow. A comparison of the thrust coefficient for LES-AL and BEM in floating applications is carried out in Apsley and Stansby (2020).

Nonetheless, whether or not these differences can be attributed to the modelling of the induction zone or the blade aerodynamics is not entirely clear. In this work, we address the question of how the general flow pattern of the inflow is influenced by the
induction zone and how it affects the turbine loads. This includes a correlation analysis (flow to load) as well as an investigation of the influence of the induction zone on the turbulent fields, which is carried out for multiple flow scenarios.

Atmospheric turbulence is a crucial factor in regular energy production because wind fluctuations influence all aspects of the turbine. Consequently, turbulence modelling is a cornerstone of wind energy research Veers et al. (2019); Kosović et al. (2025). The IEC standard specifies synthetic wind field models for emulating the effects of atmospheric turbulence. The Mann model
Mann (1994, 1998) and the Kaimal model Kaimal et al. (1972), as well as their parametrisations, are prescribed for this purpose.

As wind turbines are constantly being improved, i.e., reaching the physical limitations of the materials, the state-of-the-art development approach, based on BEM simulations, is reaching its limits. This is reflected in discrepancies between the simulated loads and the observed loads Schubert et al. (2025). In principle, the origin of such differences may lie in the already
described issues in the simulation models or inaccuracies within the turbulence prescription. For an efficient use of material and resources, as well as for ensuring the structural integrity of the turbines, it is necessary to determine the loads precisely. Therefore, improvements on both the turbulence description and the modelling assumptions are desirable.

There are various approaches to optimising turbulent fields. The recent work of Syed and Mann (2024) and Syed and Mann (2023) focuses on low-frequency, anisotropic wind fluctuations in the marine atmosphere. Ref. Syed and Mann (2024) pro-
vides a model that extends turbulence spectra to $\approx 1\ h^{-1}$ by incorporating a two-dimensional formulation for large-scale fluctuations. In Syed and Mann (2023), a Fourier-based method is presented to generate synthetic wind fields combining the two-dimensional spectral tensor from Syed and Mann (2024) for the large structures and the uniform shear model from Mann (1994) for the small scales. In the works of Kleinhans (2008); Yassin et al. (2023); Friedrich et al. (2022), the correct representation of the small-scale structures in the inertial subrange is addressed. The velocity increments on the scale of a wind turbine
and smaller are non-Gaussian distributed according to the K62 turbulence model. This property has been demonstrated for





atmospheric turbulence in various works, cf. Mücke et al. (2011). However, this phenomenon is not considered by the models prescribed in the IEC standard.

The two previous strategies for improving turbulent fields are based on physically explainable gaps in the assumptions of the models currently in use. Schubert et al. Schubert et al. (2025) have chosen a different, engineering-based approach. In their work, load measurements from a turbine are analysed regarding their damage equivalent load (DEL). It turns out that particular events, so-called bump events, which occur over time scales larger than 10 s, dominate the overall DEL. Interestingly, these large-scale events, identified in the time series of the loads, can also be found in the time series of a quantity calculated purely from the inflow wind field. Schubert et al. Schubert et al. (2025) introduced this quantity as the Center of Wind Pressure (CoWP) to describe these large-scale events. This new characteristic quantity reduces the turbulent loads to a single point in the rotor plane. A pressure-induced yaw- and tilt-moment, i.e., bending moments at the main shaft of the turbine, can be calculated based on the CoWP location. The authors observed a good agreement between the DEL from the introduced pressure-induced moments and the BEM-simulated moments.

Because these pressure-induced moments can be calculated exclusively from turbulence inflow data — independent of the wind turbine — load estimates can be obtained early in the development process. Building on this concept, Moreno et al. Moreno et al. (2024) aim to describe the dynamics of the CoWP using stochastic models, particularly the Langevin approach.

This paper extends the investigation of the CoWP, already introduced in BEM Schubert et al. (2025); Moreno et al. (2024), by analysing the effect of the simulation method on the CoWP. For doing so, three simulation approaches are compared: BEM, LES-AL, and LES-BL. The simulation models are compared under different flow scenarios, ranging from laminar to turbulent cases, thereby generalising the previous studies. By comparing the various simulation models while simultaneously relating them with the flow, it can be shown that modelling the induction zone at LES-AL results in a better correlation with the loads than with BEM. Whereas the work of Moreno et al. Moreno et al. (2024) quantitatively describes the relationship between the CoWP and wind turbine loads, it provides no one-to-one correspondence between inflow and aerodynamic response. This gap is closed in this work by introducing a calibration parameter that can be determined from a laminar simulation. The simulation settings for each approach and the selected flow scenarios are detailed in section 3. Steady inflows are analysed in section 4.1 and section 4.2. The behaviour of the turbulence, including the CoWP is analysed in section 4.3.2. Subsequently, section 4.3.3 shows how the CoWP affects a turbine in a LES simulation.

## 2 Fundamentals

The following chapter explains specific aspects of the fields used in this work, namely turbulence and numerical models. For turbulence, these are synthetic turbulence (section 2.1) and the Center of Wind Pressure (section 2.2). The numerical models are BEM (section 2.3) and CFD (section 2.4) with the sub-model AL (section 2.5).





## 2.1 Synthetic Turbulence

The use of synthetically generated turbulence to mimic the influence of real atmospheric turbulence is the procedure specified by the IEC standard IEC (2019). Although it is based only on Gaussian-distributed velocity increments, the Mann model Mann (1994, 1998) is employed in this study, as it remains prescribed in the IEC standard and is widely adopted in the literature. Despite the Gaussian statistics, it provides a controlled and computationally cheap way to reproduce key features of turbulence. These are a homogeneous field with spatial correlations, where the energy within the inertial subrange is distributed to follow the Kolmogorov -5/3 law Kolmogorov (1941). To achieve this, the turbulent fluctuations $u_{i,turb}$ are constructed in Fourier space, ensuring that the energy spectrum is correct, and are then transformed to 3D space afterwards. Only three values are required for parametrisation: A length scale $L$ to define the inertial subrange, a parameter for viscous dissipation $\epsilon^{2/3}$, and a shear distortion parameter $\Gamma$ that controls anisotropy by stretching the turbulent structures. In most cases, the Turbulent Intensity (TI) is used for the parametrising instead of the viscous dissipation as it is easier to measure and understand, cf. Larsen and Hansen (2007).

## 2.2 Center of Wind Pressure

The Center of Wind Pressure (CoWP) introduced by Schubert et al. Schubert et al. (2025) is a new characteristic quantity to describe flow structures and their influence on the loads of a wind turbine. The background to this was that certain load events (so-called "bump events") identified from operating measured data could not be realistically reproduced or explained from numerical simulations using the turbulent fields from the given standards.

The CoWP is a measure to grasp the spatial non-uniformity of the velocity field. It is described as the point in a velocity plane at which the total dynamic pressure from the velocity field acts. The formulation of Moreno et al. Moreno et al. (2024) is used in this work. The $CoWP_k$ has two components $k = [y, z]$ and is calculated from N discrete points in the velocity plane and their velocity in the main flow direction $U_x$

$$CoWP_k(t) = \frac{\sum_{i=1}^{N} k_i \cdot U_x^2(y_i, z_i, t)}{\sum_{i=1}^{N} U_x^2(y_i, z_i, t)}. \tag{1}$$

For a turbulent wind field, the CoWP is therefore a time-dependent coordinate in a plane parallel to the rotor surface, which can be determined from synthetic data or measurements. Figure 1 shows the time series of the two components, Y and Z, of the CoWP from a synthetic wind field in a) and b). Two particular times are marked by the red and green dots. Those correspond to the global maximum and minimum of the $CoWP_Z$. The instantaneous velocity planes of the wind field at those two times $t = 250\,s$ and $t = 588\,s$ are shown in c) and d). The location of the CoWP and the center of the section are marked by the red and green dots and the black crosses, respectively. The location of the $CoWP_Z$ can be explained in the velocity planes by the presence of regions with higher velocities in the upper and lower ranges, respectively. At this point, it should be briefly noted that the CoWP is relatively close to the center of the rotor plane, with amplitudes of approx. 3 m. Due to the high thrust, this offset still results in considerable bending moments on the main shaft.



In the work by Moreno et al. Moreno et al. (2024), a characterization of the dynamics of the CoWP is carried out based on the statistical properties of the signals. The Langevin approach Friedrich and Peinke (1997) is used for the characterization, i.e., by calculating the drift and diffusion values of the system. Due to the strong correlation to the bending moments at the main shaft, the dynamics of the CoWP are used for reconstructing random signals of the moments. The work shows that the combination of the CoWP and the Langevin approach allows an estimation of the loads without a simulation or even a wind field, as the loads are determined from a stochastic process. The main advantage of the stochastic reconstruction is that very long time series can be generated efficiently, which is essential for the assessment of the loads over the lifetime of the turbine.

## 2.3 Blade Element Momentum Theory

BEM theory is a fundamental analytical tool used to predict the aerodynamic performance of propellers and wind turbines. It integrates two concepts: blade element theory Froude (1878), which examines the forces on individual blade sections, and momentum theory Rankine (1865), which considers the conservation of linear and angular momentum in the flow through the rotor plane.

In BEM theory, the rotor blade is divided into numerous small elements along its length, which are assumed to be independent of each other. The local relative velocity and the angle of attack are calculated for each element based on the rotational speed and the turbulent inflow. The local lift and drag forces are determined from lookup tables for the airfoil sections. These aerodynamic forces are then used to compute the contributions to thrust and torque from each blade segment. In parallel, momentum theory is applied to account for the induced velocities in the axial and tangential directions resulting from the energy extracted by the rotor.

Since the Navier-Stokes equations are not solved in a discretised flow domain, BEM simulations are fast and widely used. At the same time, this constitutes the major drawback of BEM, since it can lead to substantial deviations from reality. Especially for the accurate modelling of complex flow phenomena near the blade tip and the blade root, as well as for unsteady aerodynamics such as dynamic stall, the differences to measurements or high-resolution models are notable. To address these issues, there are different models to correct the initial calculation, cf. Glauert (1963).

## 2.4 Computational Fluid Dynamics

In CFD, the Navier-Stokes equations are used to simulate fluids. For incompressible flows, these are

$$\nabla \cdot \mathbf{U} = 0, \tag{2}$$

$$\partial \mathbf{U}/\partial t + \left(\mathbf{U} \cdot \nabla\right)\mathbf{U} = -\nabla p + \nabla \cdot (\nu \, \nabla \mathbf{U}) + \mathbf{F}. \tag{3}$$

Whereas $\mathbf{U}$ is the velocity vector, $p$ is the kinematic pressure, and $\nu$ is the kinematic viscosity. $\mathbf{F}$ is the source term with which external forces, such as gravity, can be applied to the fluid. As proposed by Spille-Kohoff and Kaltenbach (2001) and



**Figure 1.** Time series of the CoWP a) and b). Slice of a field generated by a synthetic turbulence model in c) and d). The CoWP locations are shown by the red and green dots. The center of the slice is shown by the black cross.





Gilling and Sørensen (2011), this source term can also be used for a turbulent inflow inside the domain. For this purpose, the fluctuations from the wind field $u_{i,turb}$ are considered as accelerations of the background velocity, which is then converted into
155 a force

$$F_{i,c} = \frac{1}{2} A_c \left( U + \frac{1}{2} u_{i,turb} \right) u_{i,turb}. \tag{4}$$

### 2.5 Blade resolved and Actuator Line wind turbine representation

The most obvious representation of a wind turbine in CFD is blade resolved (LES-BL). For this, the exact geometry of the wind turbine is resolved by the numerical grid. This requires a large number of small cells around the blades in order to be
able to capture all aerodynamic effects. Due to the small cells, a small time step is required for the simulation as well. The combination of many cells and a small time step makes blade resolved simulations computationally intensive.

The Actuator Line Method (LES-AL) introduced by Sørensen and Kock (1995) is a computational technique used in CFD to simulate wind turbine aerodynamics efficiently. Instead of modelling the full geometric complexity of turbine blades, LES-AL represents each blade as a line of discrete force elements distributed along its span. These elements apply forces to the flow field
through the source terms **F** in Eq. 3, replicating the aerodynamic effects of the blades without the need for detailed geometric resolution.

In LES-AL, the forces are calculated based on local flow conditions from the CFD field and airfoil characteristics from lookup tables. The force determination for the LES-AL is based on the same lookup tables as for BEM methods. To mitigate singularities and numerical instabilities, the body force vector is distributed over the flow field using a Gaussian function Sorensen and
170 Shen (2002).

This approach allows for the capture of essential aerodynamic interactions between the turbine and the surrounding flow field, including wake formation and evolution, while significantly reducing computational costs compared to fully resolved LES-BL simulations, by modelling the actual airfoil flow interaction.

### 2.6 Comparison of the different methods

When developing a new wind turbine, various tools for load prediction are available. They differ in model complexity and, consequently, in the computational effort required to simulate a specific load case. The crucial question is what level of detail is required for the load prediction for the specific components of a wind turbine.

Table 1 shows a comparison of various existing tools for load prediction. BEM, LES-AL and LES-BL are frequently used and well established in research. Their respective advantages and disadvantages are commonly known and well documented.
The newly introduced CoWP differs from previously described models, as it has so far been presented exclusively using BEM simulations. So it remains unclear whether the concept can also be generalised for high-resolution LES simulations. Furthermore, the signals are normalised in both papers, and it remains to be clarified how CoWP can be converted into a load signal.





**Table 1.** Overview of different load prediction tools.

| Simulation model | BEM | LES-AL | LES-BL | CoWP |
|---|---|---|---|---|
| Accessible flow field | - | ++ | ++ | - |
| Modelling of multiple turbines | - | ++ | + | - |
| Airfoil aerodynamics | - | - | ++ | - |
| Root/Tip vortex representation | - | + | ++ | - |
| Dynamic Stall | - | - | ++ | - |
| Calculation time | + | - | - - | ++ |
| Bending moment on the main shaft | + | + | + | + |

## 3 Methodology

Here, a detailed presentation of numerical setup is given. It starts with the selected turbine and the operating point (section 3.1). Followed by parametrisation for the BEM simulation (section 3.2) and the CFD simulation (section 3.3) in terms of the solver, the grid, the numerical schemes and turbulence models. It ends with a physical derivation of the Load center (section 3.5) and a description of the selected flow scenarios (section 3.6).

### 3.1 Turbine Setting

The investigation in this work is carried out with the NREL 5MW reference turbine Jonkman (2009). With a diameter of 126 m, this turbine represents the state-of-the-art turbine. To neglect all periodic loads, a very simplified rotor is represented, i.e., the rotor is not tilted, the blade has no cone angle, and a constant pitch angle. Additionally, there is no tower (similar to Dose et al. (2018)). The turbine is operated in rated conditions ($U_x$ = 11.4 m/s), with a constant rotor speed of 12.1 rpm.

### 3.2 BEM Setup

The BEM simulations are performed with the open-source tool OpenFAST v2.5 with the provided repository for the NREL 5 MW reference turbine Jonkman et al. (2021). The controller, gravity, ground effect, tower effects, and dynamic stall model are turned off to model the same setup as in CFD. The blade pitch angle as well as the rotational speed are defined as constant, with values of 0° and 12.1 rpm, respectively.

### 3.3 CFD Setup

The CFD simulation is carried out with the open-source toolbox OpenFOAM v2306 OpenCFD (2023). The incompressible unsteady solver pimpleFOAM Greenshields and Weller (2022) is used, which uses a combination of the PISO Issa (1986) and SIMPLE Patankar and Spalding (1983) algorithms for pressure–velocity coupling. A second-order Backward scheme is used for the time derivative, and a second-order Linear Upwind scheme is used for the convective term.

The turbulence is modelled with the standard Smagorinsky Smagorinsky (1963) subgrid scale model for the LES-AL case. For



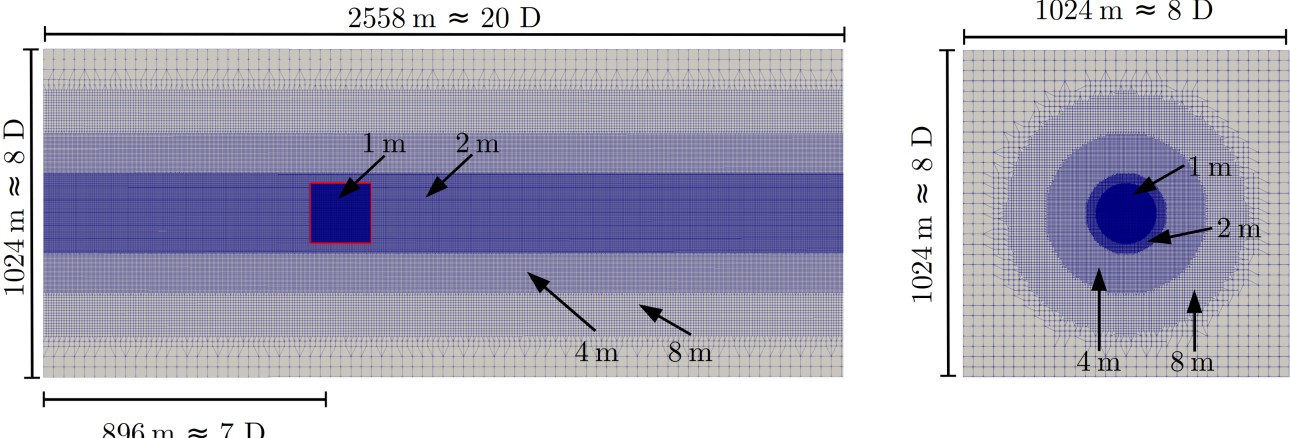

**Figure 2.** Cutting slices of the grid.

the LES-BL case, a Delayed Detached Eddy Simulation is used Gritskevich et al. (2012). This is a hybrid between the k-omega
SST model Menter et al. (2003) near the wall and a standard Smagorinsky model Smagorinsky (1963) for the farfield. This
ensures that the flow in the induction zone is computed using the same subgrid models.

### 3.3.1 Mesh settings

The same base mesh is used for all LES simulations (LES-AL and LES-BL for the three flow scenarios), which is shown in
figure 2. For the LES-BL simulations, there is an additional rotor region with the blade meshes and the hub. The simulation
domain has a length of $2558\ m\ (\approx 20\ D)$ and a width/height of $1024\ m\ (\approx 8\ D)$. In the base mesh, all cells are quads with an
aspect ratio of one. In the area of the rotor, as well as the direct near-wake, the cells have a resolution of $1\ m$. Over the entire
length of the domain, there is a cylindrical refinement zone with a diameter of $240\ m\ (\approx 2\ D)$ and a resolution of $2\ m$. Further
outwards, the cells become coarser in other refinement zones, resulting in a total cell count of 27.2 million cells. A grid study
is attached in Appendix A.

### 3.3.2 Actuator Line Model

The actuator line implementation in OpenFOAM used in this work employs the version by Bachant et al. (2016, 2024). There
the required airfoil lookup tables for the 5 MW reference turbine are provided in the tutorials. The turbine is set up without a
tower by commenting out this section. For modelling the tip and root losses, the Glauert model Glauert (1963) is used. Along
the span, 57 points per blade are used. To extract the sectional forces, both the blade performance and element performance
options are enabled.





### 3.3.3 Blade resolved settings

The blade mesh is created with the in-house blade meshing tool, blade block mesher Schmidt et al. (2012). In this grid gen-
eration tool, several structured 2D airfoil sections are connected along the span. The blade mesh of this paper is the same as
in the work of Dose et al. (2018) and Höning et al. (2024). It is a C-mesh topology with a resolution of 300 cells chordwise
and 40 cells normal to the wall, with a growth ratio of 1.2. Along the span resolution, 260 cells are used, totalling 3.56 million
cells per blade. The first cell resolution is chosen for a high-Re approach with wall functions, where the majority of the cells
are within $30 < y+ < 70$. The base mesh with blade and rotor mesh combined has a total of 44.6 million cells.

### 3.4 Calculation time

The varying complexity of the different models results in a significantly different calculation effort. The BEM simulations are
carried out on a local workstation. A case of 200 s simulation time takes approximately 75 s (wall time). In other words, to
simulate 1 s with one processor, 2.67 CPUs are required (wall time divided by number of processors; assuming BEM runs
serial). For the LES-AL simulation, this corresponds to 45,000 CPUs per second (parallel on 128 cores), and for a LES-BL
simulation, 1,500,000 CPUs per second (parallel on 256 cores). To summarise, this means that an LES-AL simulation is 16,800
times more costly than BEM, and a LES-BL simulation is even 561,000 times more expensive than BEM.

### 3.5 Center of Pressure and Load Center

In order to explain the methodology used in this paper in detail, we will briefly repeat the Center of Pressure (CoP). This is
a well-known and established concept in fluid dynamics, cf. Anderson Jr (2016). It is the location from which a point force
has the same effect on an object as the pressure forces acting on the surface. The CoP location is often used to describe the
stability of sailing boats, aircraft, or cars. It is calculated by setting up a matrix equation with the aerodynamic forces $\mathbf{F_{aero}}$
and aerodynamic moments $\mathbf{M_{aero}}$ and solving for the CoP

$$\mathbf{F_{aero}} \times \mathbf{CoP} - \mathbf{M_{aero}} = 0. \tag{5}$$

This makes the CoP in our case a turbine-specific variable that is calculated from the pressure distribution in response to the
flow around the turbine.

Schubert et al. Schubert et al. (2025) applies the idea of the CoP to a flow field by replacing pressure with the velocity squared.
This makes the CoWP a pure flow quantity that is independent of any object.

With the same motivation as with the CoWP, the Load center a new quantity introduced now. As there is no resolved geometry
in the BEM and LES-AL models, the CoP can not be calculated. Instead, there are sectional forces of the blade segments. For
a given timestep of a BEM or LES-AL simulation, the location of each blade segment and the corresponding thrust force $F_x$
are known. Instead of a velocity plane at the CoWP, a sparsely filled plane with the thrust forces is used for the Load center

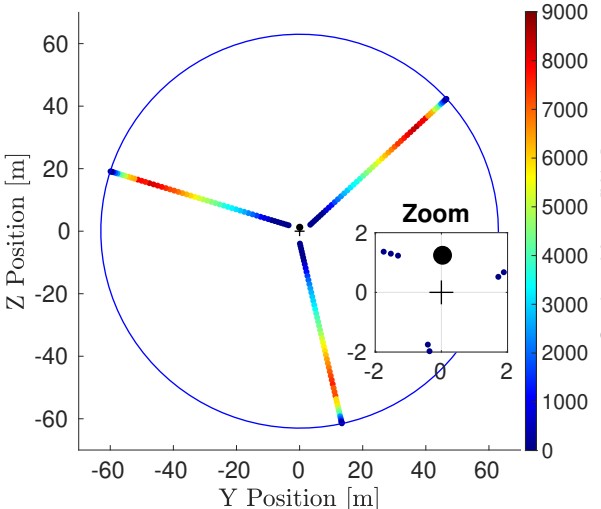

**Figure 3.** Force array for the Load center estimation of an LES-AL Simulation, showing the sectional forces. The center of the rotor is shown by a black cross, and the rotor area is marked by a blue line. A big black dot shows the Load center.

calculation (shown in figure 3). The Load center is then calculated in the same way as the CoWP.

$$Load\ center_k(t) = \frac{\sum_{i=1}^{N} k_i \cdot F_x^2(y_i, z_i, t)}{\sum_{i=1}^{N} F_x^2(y_i, z_i, t)}. \tag{6}$$

In summary, there are three quantities with the unit metre, all of which represent a distance from the rotor center. Therefore, these quantities are ideally suited for comparison with each other. The CoP and Load center both refer to the turbine-specific blade forces. So for the sake of simplicity, this paper always refers to the Load center when speaking about loads, even if the CoP is meant for the LES-BL cases. In table 2 there is a summary of the given quantities.

**Table 2.** Overview of the different location quantities.

| Name | Calculated from | Used for |
|------|-----------------|----------|
| CoP | Pressure distribution [Pa] | LES-BL |
| CoWP | Velocity field [m/s] | all cases |
| Load center | Sectional forces [N] | BEM/LES-AL |

### 3.6 Flow scenarios

Three flow scenarios are investigated (see figure 4):

1. A uniform laminar flow as proof of concept and to determine the basic uncertainty of the models




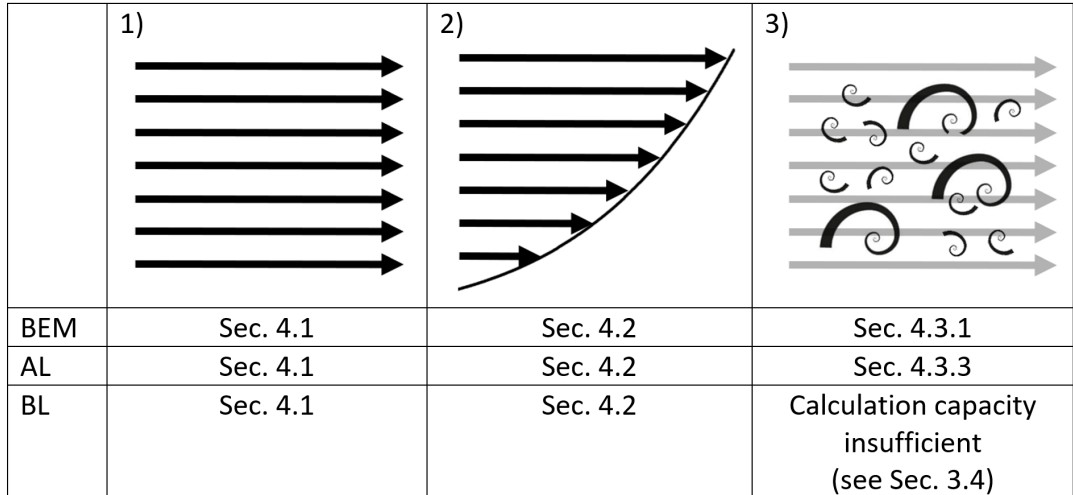

| | 1) | 2) | 3) |
|---|---|---|---|
| BEM | Sec. 4.1 | Sec. 4.2 | Sec. 4.3.1 |
| AL | Sec. 4.1 | Sec. 4.2 | Sec. 4.3.3 |
| BL | Sec. 4.1 | Sec. 4.2 | Calculation capacity insufficient (see Sec. 3.4) |

**Figure 4.** Schematic illustration of the flow scenarios. 1) Uniform laminar 2) Laminar shear flow 3) Turbulent. Additionally, the corresponding sections are shown below the pictures.

2. A laminar shear flow to determine the differences between the methods and how the shear profile interacts with the turbine. A power law profile with an exponent of $\alpha = 0.143$ is used, which is a typical value for an offshore location Hsu et al. (1994). The hub height of the turbine is used as the reference height.

3. A turbulent flow to determine a realistic case. The Mann model is used for generating the turbulent wind field (see section 2.1). The field is parametrized by $L = 126$ m (= 1D) and TI = 5%. To simplify the analysis, no shear is used ($\Gamma = 0$), as the Taylor hypothesis for frozen turbulence Taylor (1938) can thus be applied.

    The field should have a resolution of 2 m in each spatial direction to be consistent with the recommendation of Troldborg et al. (2014). Furthermore, the wind field should fill the entire LES domain and enable a simulation of 10 minutes – the usual investigation interval in the wind energy field. The average speed and spatial resolution result in a temporal resolution of 0.175 s. The required wind field must therefore have dimensions of 6860 m x 1024 m x 1024 m (3430 x 512 x 512 points). To create such a field with $\approx$ 900 mio points, the turbulence generator introduced by Liew et al. (2023); Liew (2022) is used.

## 4 Results and discussion

In the following section, the results are presented in the order of the flow scenarios from section 3.6. Starting with the laminar flow in section 4.1. The shear flow and the calibration factor are presented in section 4.2. Then the turbulent inflow with BEM





is shown in section 4.3.1. The turbulent characterisation in an empty box is done in section 4.3.2. And finally, the turbulent LES-AL case is in section 4.3.3.

## 4.1 Uniform laminar flow

In the uniform laminar flow case, the inlet velocity is the same everywhere. Consequently, the position of the CoWP is in the center of the rotor surface (derived Eq. (1). The course of the Load center in the BEM simulation is trivial, with zero in Y and Z. Whereas the Load centers for LES-AL and LES-BL deviate slightly from the center of the rotor, shown in figure 5. The standard deviation is $7.20\ 10^{-3} m$ in Y and Z for the LES-AL and $2.06\ 10^{-2} m$ in Y and $1.99\ 10^{-2} m$ in Z for the LES-BL. As can be seen in section 4.2, 4.3.1 and 4.3.3, these fluctuations are one order of magnitude smaller than for the shear and turbulent case. Despite the minor deviation compared to the other flow cases, the causes are being investigated in order to understand the intrinsic properties of the models.

In the LES-AL simulation, the fluctuations appear at the 3 P frequency and can therefore be attributed to interpolation errors between the Cartesian grid and the rotational blades. Those errors are a well-known characteristic of LES-AL simulations, which occur when the body forces are applied to the portion of the domain where the blades are, cf. Churchfield et al. (2017). Finally, we come to the LES-BL case. To explain the fluctuation there, we need the Q-criterion (Davidson (2015)) and the sectional forces on the blade. Figure 7 visualises the isosurfaces of the Q-criterion around the rotor for the LES-AL in a) and the LES-BL b). In the LES-AL, only the three helical tip vortices and a glimpse of root vortices can be recognised. In the LES-BL case, many small detached vortex structures appear near the blade root, due to the turbine blade's cylindrical cross-section up to a radius of 8.3 m. The flow around this cross-section is typically detached, and each blade is strongly influenced by the wake of the others, so no periodic vortex patterns are formed.

Figures 8 a) and b) show the time-averaged and time-dependent course of the sectional blade forces for the LES-BL case. Figure 8 c) shows the relative standard deviation to the mean value over the blade length. In general, the sectional forces confirm the conclusions from the Q-criterion analysis. The forces near the blade root, where a cylindrical cross-section is present, fluctuate strongly, with a standard deviation of over 10%. Further outwards, there is a transition segment to an airfoil (at half of the blade length), from which the forces are more or less constant with a relative standard deviation of less than 0.2%.

The strong fluctuations in the forces at the blade roots cause the Load center does not always coincide with the centre of the rotor surface (figure 5). This offset, in the order of few centimeters, results from the randomness of flow being detached/attached to the three rotor blades.

## 4.2 Laminar shear flow

Since there is a velocity gradient in the inlet for the shear case, the position of the CoWP is not straightforward. The calculated $CoWP_Z$ is 3.39 m and $CoWP_Y = 0$ for the inlet boundary condition of the simulations.

Figure 9 shows the time series of the Load centers for the different simulation methods. In contrast to the laminar case, the Load centers fluctuate periodically for all methods. Due to the velocity gradient of the shear flow, the load on each blade varies



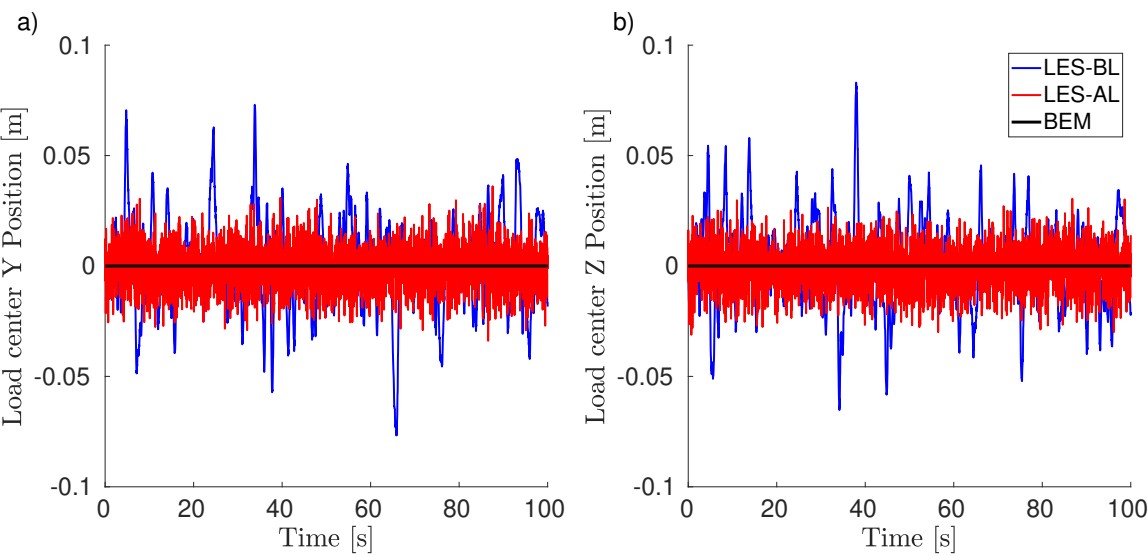

**Figure 5.** Time series of the Load center for the three simulation methods with laminar inflow. Y component in a) and Z component in b).

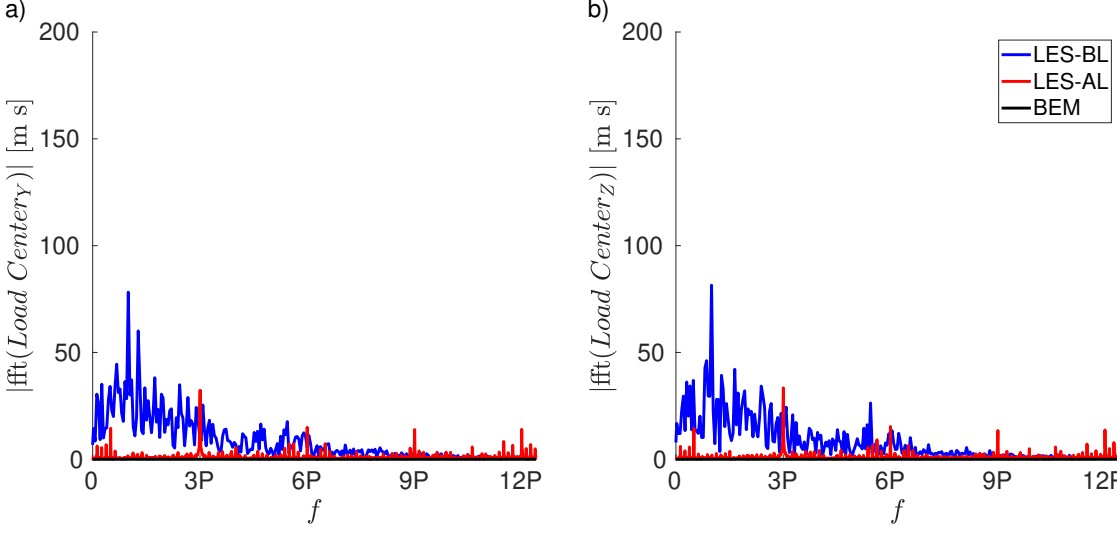

**Figure 6.** FFT of the Load centers for the three simulations methods for the uniform laminar case. Y component in a) and Z component in b) (3 P = 0.605 Hz).

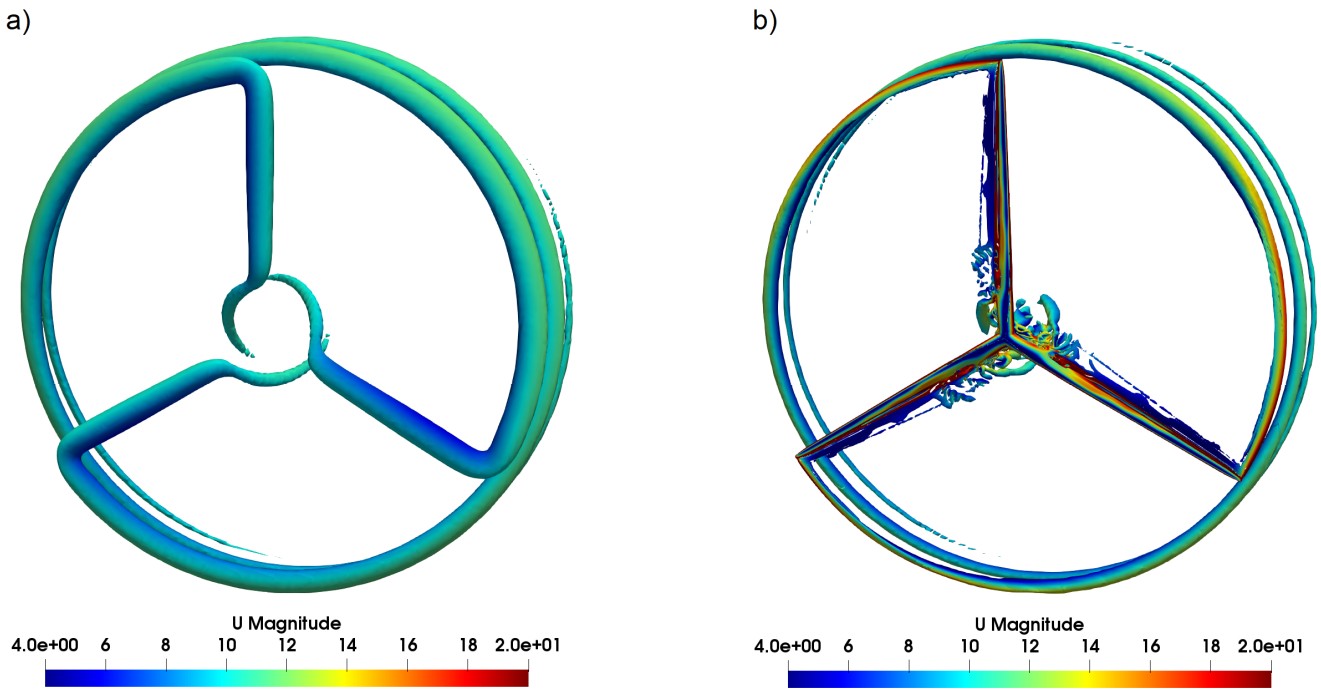

**Figure 7.** Isosurfaces of the Q-criterion for a) the LES-AL (Q = $0.2\ s^{-2}$) and b) the LES-BL (Q = $1.2\ s^{-2}$) visualised with the flow velocity.

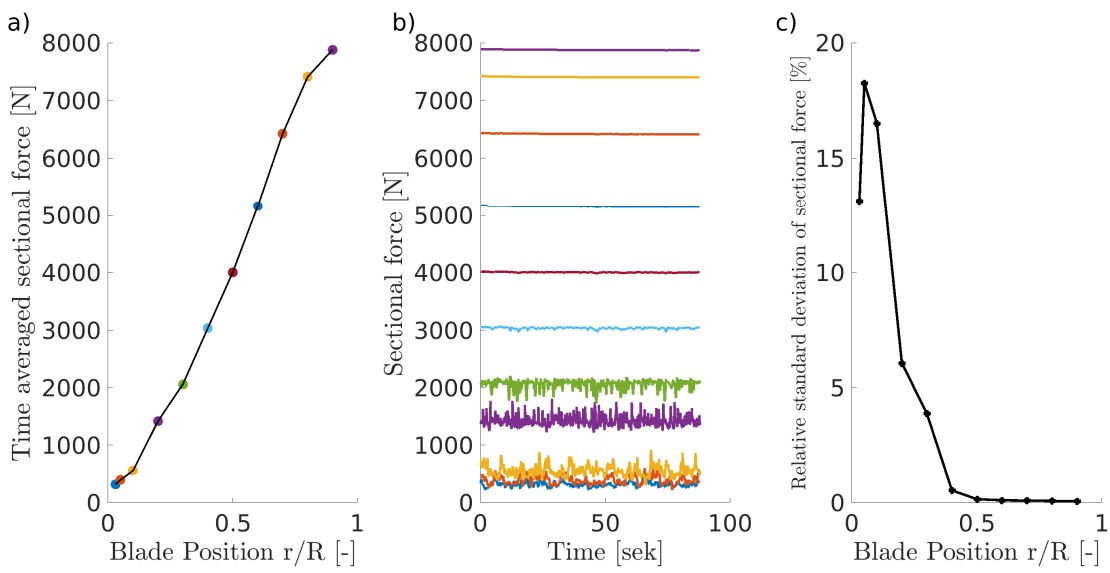

**Figure 8.** Sectional forces of a blade resolved simulation with laminar inflow. Time-averaged sectional forces over blade radius a). Time-series of the sectional forces b). Relative standard deviation of the sectional forces over blade radius.





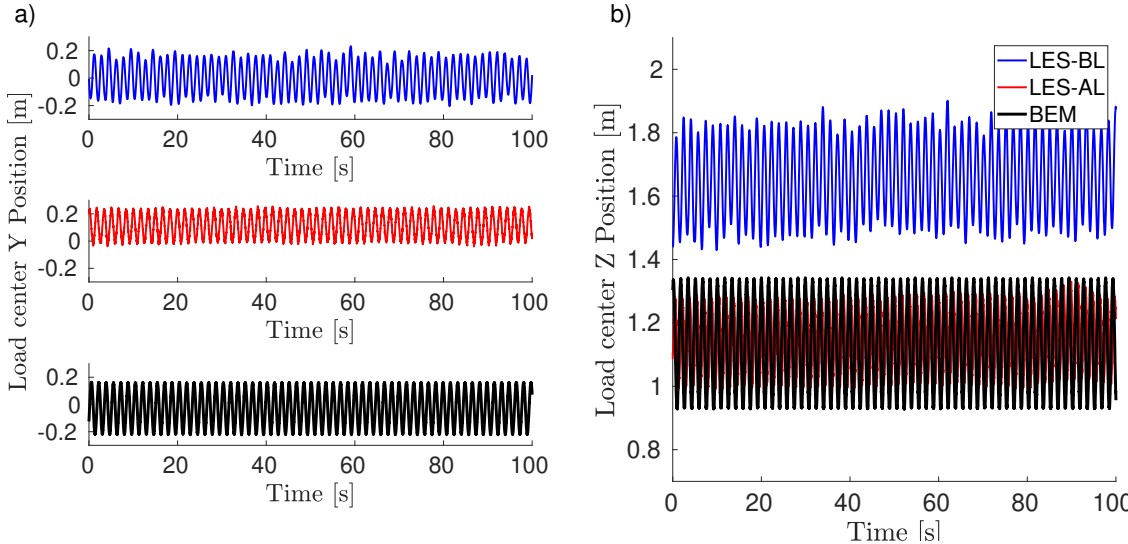

**Figure 9.** Time series of the Load center for the three simulation methods with shear flow. Y component in a) and Z component in b).

during each revolution. Because the blades are geometrically coupled, the Load center rises when two blades are above the nacelle and falls when two blades are below it. The same applies to the Y component, corresponding to the blade position and the associated Load center. As can be seen in figure 10, the 3 P frequency of revolution (0.605 Hz) is the dominant one for all simulation methods.

The mean Load center in the Z component is similar for BEM and LES-AL with 1.13 m and 1.16 m. The Load center for the LES-BL simulation is with 1.66 m substantially higher than the simulation models, where flow around the blades is not resolved. Now that we have the Load centers from the actual forces and the CoWP from the boundary condition, we can determine the relationship between them. This relationship will later be used in section 4.3 as a calibration factor for the turbulent case. As already mentioned, in the introduction, this relationship was previously unclear. The mean values for the

Load centers as well as the ratio between the Load centers and the $CoWP_Z$ are given in table 3.

Like the $CoWP_Y$, the mean Load center in the Y component is zero for BEM and LES-BL (figure 9 a). For LES-AL the mean Load center is 0.12 m. The shift is related to the direction of rotation of the turbine, as shown in Appendix B.

**Table 3.** Load center and calibration factor for the laminar shear case.

| Simulation model | $Load\ center_Z$ | $Load\ center_Z$ / $CoWP_Z$ |
|---|---|---|
| BEM | 1.13 m | 0.334 |
| LES-AL | 1.16 m | 0.341 |
| LES-BL | 1.66 m | 0.489 |





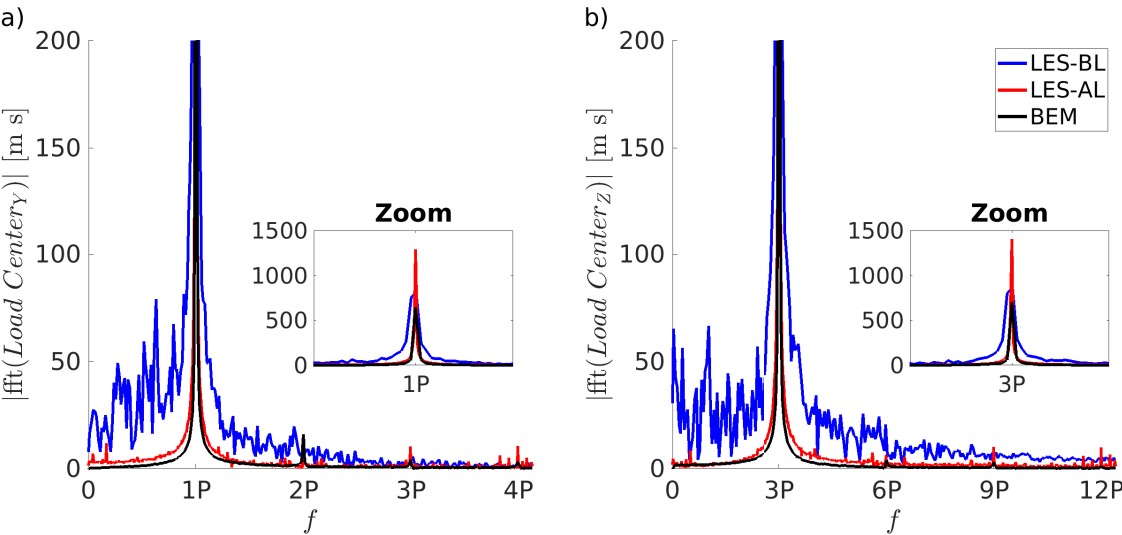

**Figure 10.** FFT of the Load centers for the three simulations methods with shear inflow. Y component in a) and Z component in b) (3 P = 0.605 Hz).

## 4.3 Turbulent inflow

This section deals with simulations involving turbulent inflow. In section 4.3.1, the turbulent wind field is used as inflow for
BEM. As already indicated in the introduction, LES involves a spatial-temporal development of turbulence. Therefore, section
4.3.2 first examines the turbulent field in a simulation without a turbine, and section 4.3.3 then examines it with a turbine.

### 4.3.1 Turbulent case in BEM

The time-series of the CoWP from the synthetic inflow field as well as the Load center from the BEM simulation is shown
in figure 11. As with the laminar shear case from section 4.2, the amplitude of the load center is substantially lower than the
CoWP. Furthermore, the Load center signal exhibits many fluctuations. Similar observations were described in the work of
Moreno et al. (2024). In that work, the load component was filtered using a low-pass filter and all signals were normalised to a
standard deviation of one.

In the present work, the load center is also filtered using a low-pass filter with a cutoff frequency of 0.660 Hz ($\approx$110% of the
3 P frequency). However, a different method instead of normalisation is chosen here. The value of the CoWP is corrected by
multiplication with the ratio (Load center/$CoWP_Z$) under laminar conditions, given in section 4.2, table 3. Figure 12 shows
the time series of the filtered Load center and the rescaled CoWP. Filtering and scaling indicate a good correlation between the
Load center and the CoWP with a Pearson correlation coefficient of 0.814. Figure 13 shows the statistical analysis of the time
series of the scaled Load center and the filtered CoWP. Both in the histogram in a) and the energy spectrum in b), the load and
flow variables have similar properties.

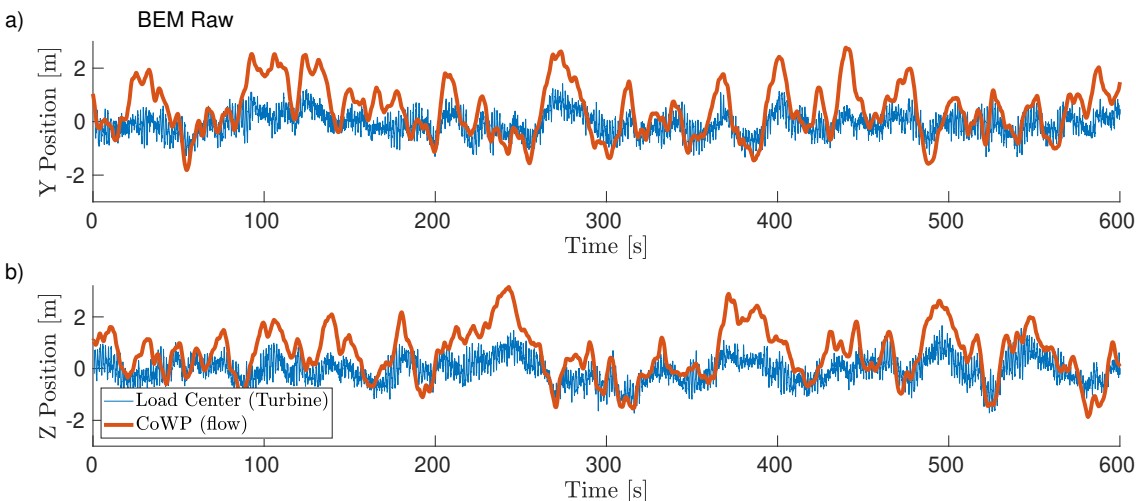

**Figure 11.** Time-series of the CoWP from the synthetic inflow field and the Load center of the BEM simulation. Y component in a) and Z component in b).

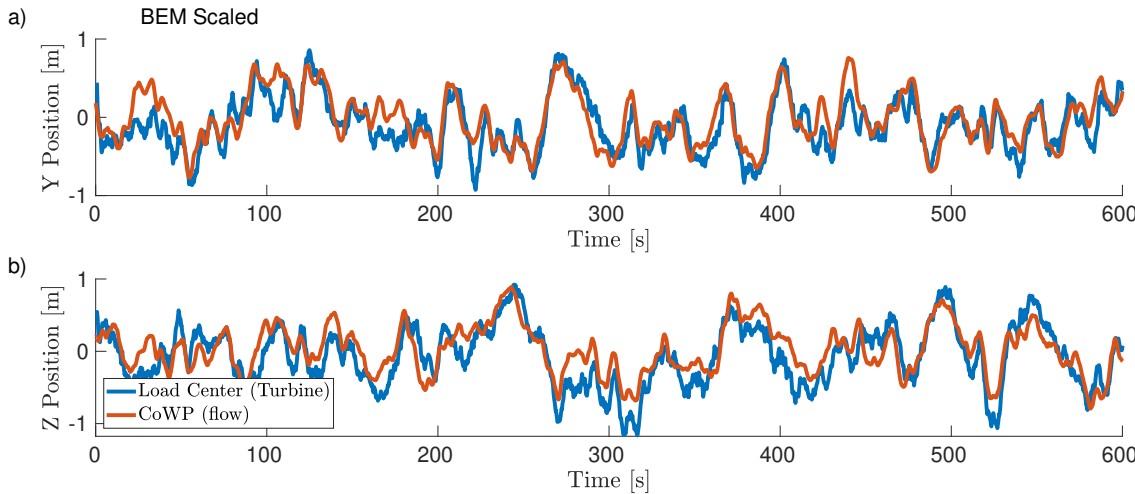

**Figure 12.** Time-series of the scaled CoWP from the synthetic inflow field and the filtered Load center of the BEM simulation. Y component in a) and Z component in b). $r_{Pearson} = 0.814$.





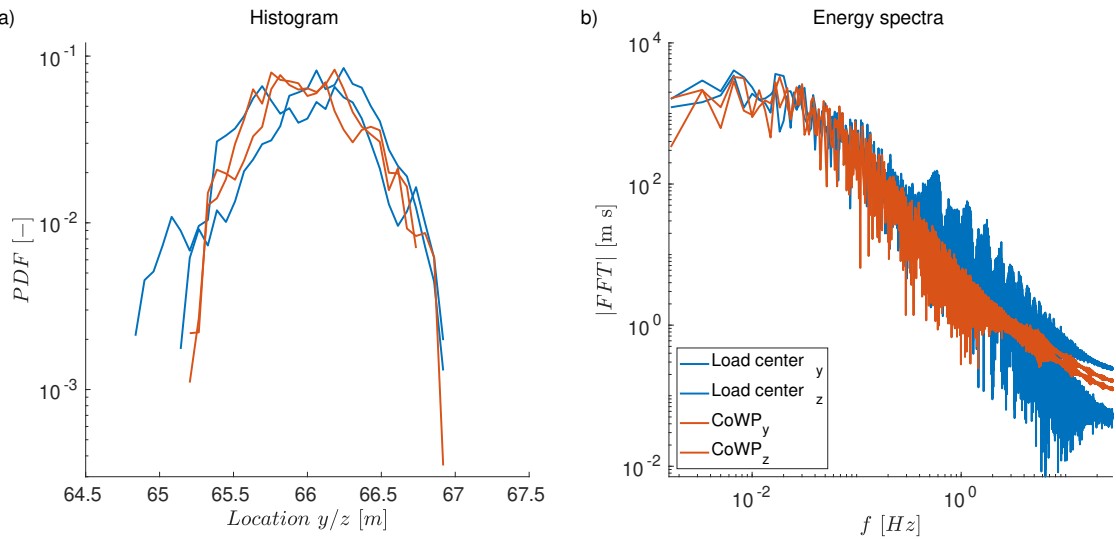

**Figure 13.** Statistical analysis histogram in a) and energy spectra in b) of the scaled CoWP and the filtered Load center for the BEM simulation (figure 12).

### 4.3.2 Turbulence characterisation in LES

Before discussing the results of the LES-AL modelled turbine under turbulent inflow, a characterisation of the flow must be carried out first. Intermittency is an intrinsic property of turbulence. As showed by Bock et al. (2024) a realistic representation of a turbulent flow can only be achieved if these characteristics have been verified. Furthermore, the turbulence interacts with the induction zone and the blades of the turbine. In order to distinguish and evaluate these two influences on the flow, a simulation without a turbine is carried out first and the characterisation proposed by Bock et al. (2024) is performed. Afterwards, a comparison of the flow with a turbine is conducted.

Figure 14 shows the characterisation of the turbulence at different downstream positions. The standard quantities, turbulence intensity (TI) and the energy spectrum are shown in a) and b). Since there is no turbulence production, it is a case of decaying turbulence. As usual with a turbulent inflow, $TI_x$ is lower than $TI_y$ and $TI_z$ after the inflow and increases within the first 100 metres Gilling and Sørensen (2011); Bock et al. (2024); Keck et al. (2014). As expected, there are negligible changes in the energy spectra.

Now that two fundamental properties of a decaying turbulent flow have been confirmed, we will examine the higher orders of the two-point statistics in figure 14 c) and d). In c) the shape parameter $\lambda^2$, which quantifies intermittency, of the increment statistics over the increment size $\tau$ is presented. For small increments, the shape parameter is $> 0$ and thus exhibits non-Gaussian, or intermittent, properties of the increment statistics. The intermittency parameter $\mu$ can be determined from $\lambda(\tau)^2$ according to the K62 turbulence model Kolmogorov (1962); Obukhov (1962); Chilla et al. (1996). A more detailed description of this method is given in Bock et al. (2024). The downstream development of the intermittency parameter is shown in figure





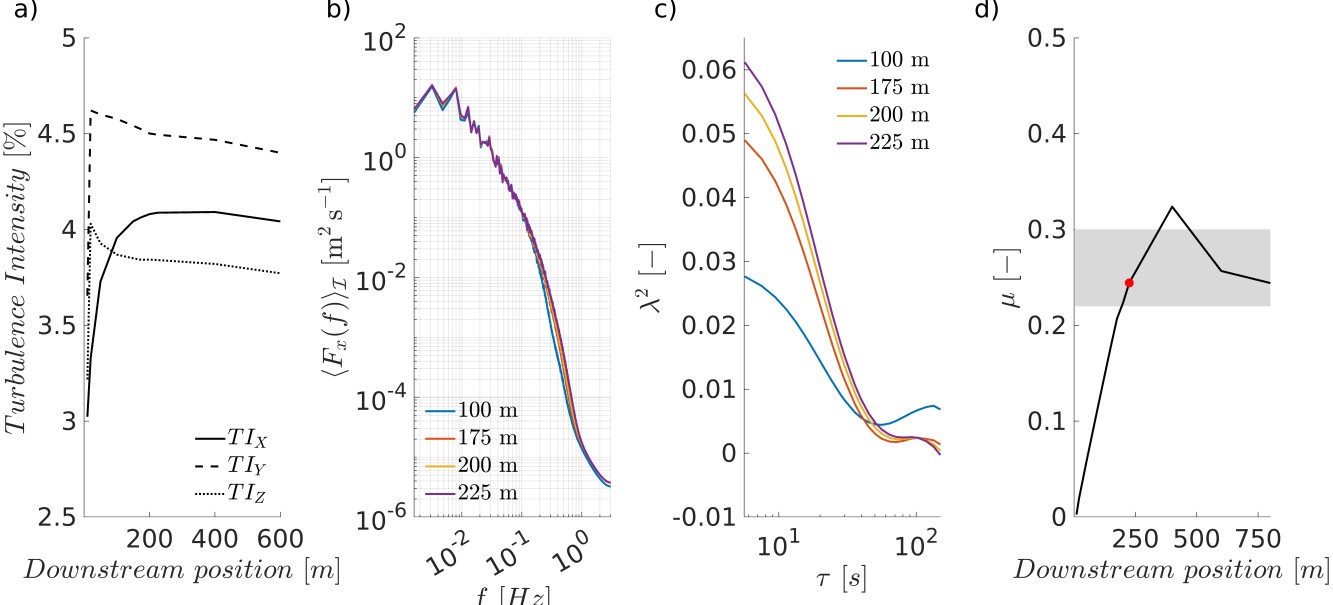

**Figure 14.** Analysis of turbulence without a turbine. Turbulence intensity over downstream position a) Energy spectrum of different downstream positions b) Shape parameters of the two-point statistics over the increment size c). Downstream development of the intermittency parameter $\mu$.

14 d). The range of the intermittency parameter for ideal turbulence in accordance with Arneodo et al. (1996) is shown as a grey area. Furthermore, the distance between the inflow and the turbine from section 4.3.3 is marked as a red dot. Overall,
the behaviour of the turbulence in LES is consistent with the results from Bock et al. (2024), which means that a realistic intermittency state is present.

Figure 15 shows the time series of the calculated components $CoWP_Y$ and $CoWP_Z$ from the wind field at different downstream positions. The input field before injection is shown in black and in different colours for the different downstream positions in LES in a) and b). There is reduction in the amplitude from the beginning on in LES compared to the inflow. After
365 that, there are further adjustments within the first 100 m (represented by the dotted lines) and essentially no changes between 150 m and 225 m. Nevertheless, the LES reproduces the basic CoWP pattern.

Similar to the TI (see figure 14 a), the CoWP in the LES changes after the inflow in the domain. This change mainly occurs within the first 150 m. Subsequently, the course of the CoWP changes only very little. A study was carried out in Appendix C to analyse the reasoning behind the deviations between inflow field and LES in more detail.

**4.3.3 Turbulent inflow with LES-AL**

The same inflow field from section 4.3.2 is used in the next step for a simulation with a LES-AL modelled wind turbine. Before the loads are considered, the flow in front of the turbine is analysed and compared with the empty simulation.

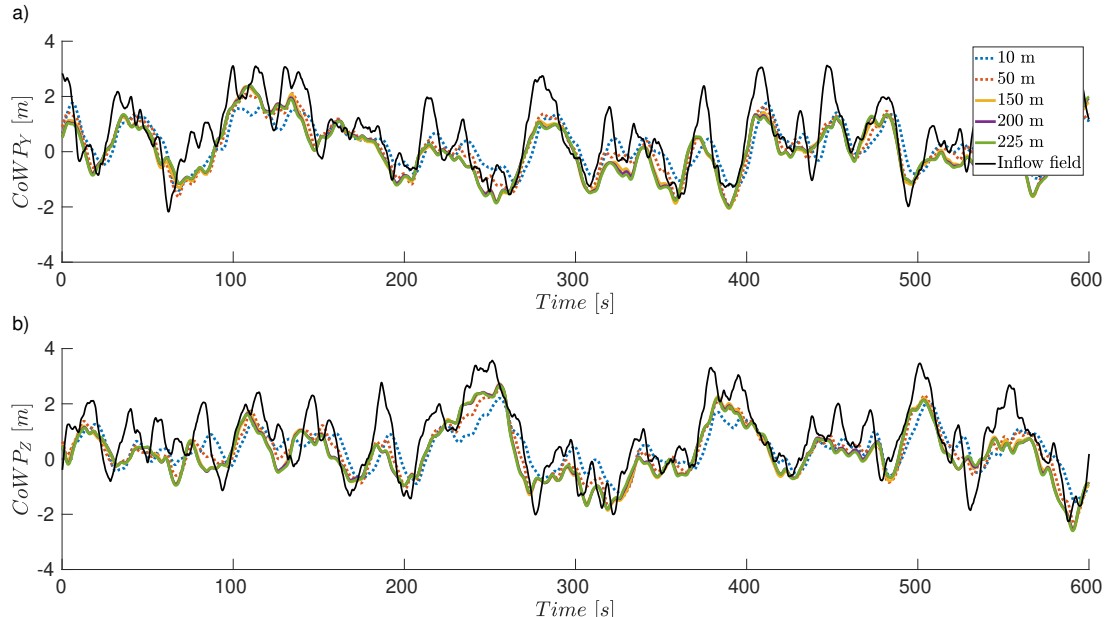

**Figure 15.** Time-series of the CoWP in the inflow field (black) and at different downstream positions in an empty box LES (coloured). Y component in a) and Z component in b).

Figure 16 shows the energy spectra for a domain with turbine in red and for an empty domain in black in the LES at different positions. At 100 m, the energy spectra are essentially the same. At 150 m, the spectra at the low frequencies are also the same. At the higher frequencies, there is a peak at 0.605 Hz, which corresponds to the 3 P rotation frequency and is due to the periodic fluctuation caused by the rotating blades. At 200 m and 225 m, the energy at this frequency continues to increase and higher harmonics of this frequency arise. However, no difference can be seen between the simulations with and without a turbine in the low-frequency amplitudes. This suggests that the large-scale structures that dominate the CoWP are only slightly influenced by the interaction with the turbine.

The time series for the CoWP in the LES is shown 200 m downstream the inflow with and without turbine in figure 17 a) and b). The absolute differences between the simulation with and without turbine is shown in c). The deviation between the simulation with and without the turbine varies ±0.6 m with a mean in both directions below 0.1 m. So the interaction of the turbine with the incoming flow is at some times with a deviation of 0.5 m quite relevant, since the absolute CoWP offsets from the rotor center are only one to two meters.

The FFT of the CoWP from figure 17 with and without turbine, presented in figure 18. It indicates that the 3 P frequency, which appears in LES-AL case due to the resolved induction of the rotating blades, adds additional noise to the CoWP for a simulation with turbine. As can be seen from the energy spectra and the CoWP, the blockage caused by the rotating blades has an influence and possibly an interaction with the turbulence. This raises the question whether the rotor position influences the





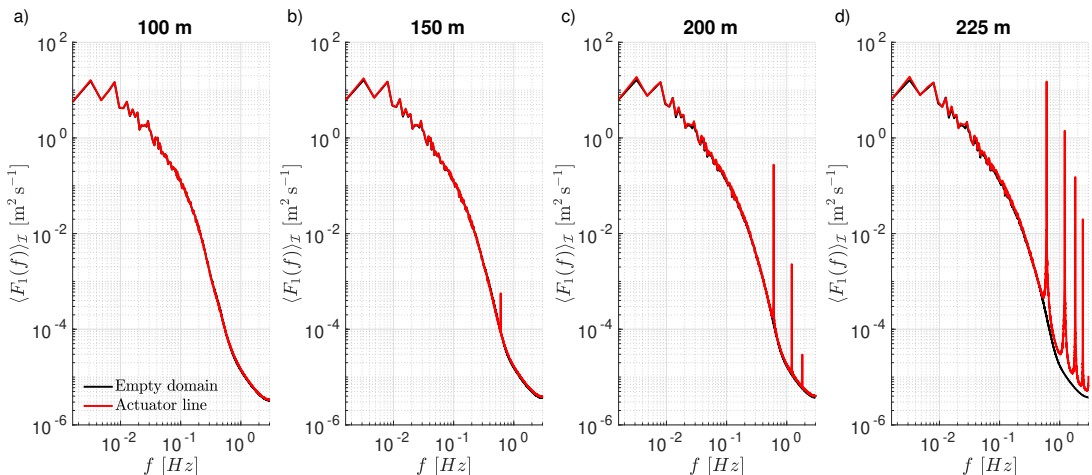

**Figure 16.** Energy spectra of the LES flow field at different downstream positions.

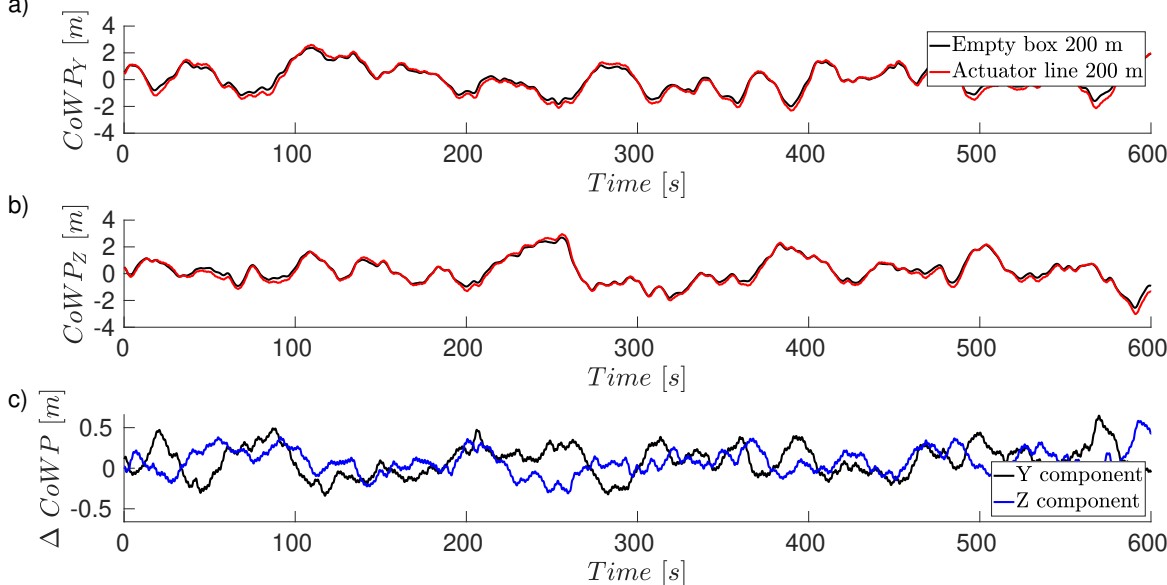

**Figure 17.** Time-series of the CoWP 200 m downstream the inflow without turbine (black) and with LES-AL modelled turbine (red) in a) and b). Absolute difference between the simulation with and without turbine in c).

loads.





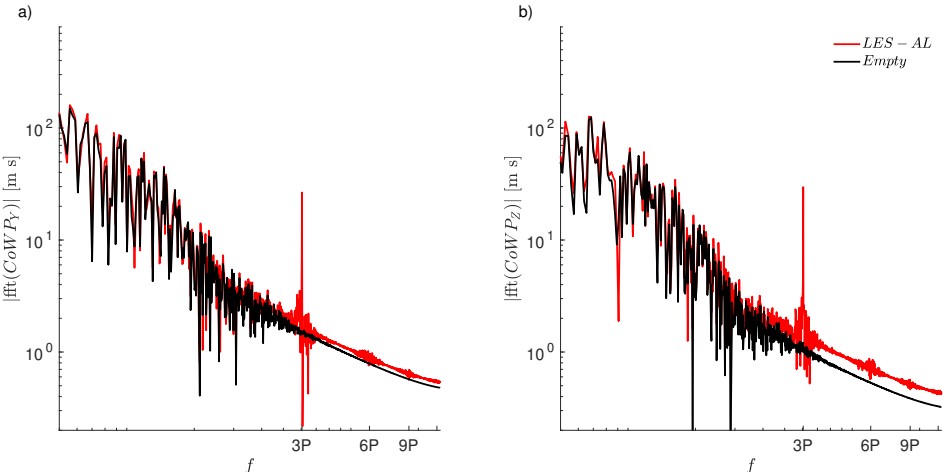

**Figure 18.** FFT of the CoWP in LES 200 m downstream the inflow (figure 17) (3 P = 0.605 Hz).

Figure 19 shows the time series of the CoWP from LES in the rotor plane and the Load center for the LES-AL simulation. As with the BEM simulation (figure 11), the load signal is noisy and the maxima of the CoWP exceed the peaks of the Load center. As in section 4.3.1, the load signal is filtered and the CoWP signal is rescaled with the calibration parameter from section 4.2, table 3 (figure 20). The correlation between the CoWP and Load center in LES-AL is even better than BEM, with

395 a Pearson correlation coefficient of 0.908. This difference arises because the flow field for CoWP calculation from the LES is actually the one that hits the turbine, whereas in the correlation from section 4.3.1, the inflow wind field is slightly modified by the BEM simulation's induction model. The histograms and energy spectra also fit, see figure 21. As already shown in the investigation of the influence of the turbine, there are influences of the rotation in the form of peaks at the 3 P frequency in both the Load center and CoWP spectrum (3 P and multiples shown by dashed lines).

**5 Conclusions and Outlook**

In this work, three wind turbine models with different fidelities were compared in terms of their correlation to the load prediction from the CoWP. The CoWP itself is a new quantity purely extracted from the inflow wind field, and therefore does not contain any information about the turbine or the local blade aerodynamics. Thus, two main questions had to be answered: Firstly, how can the CoWP be converted into a load signal to be used in the development process of a turbine. And secondly,

whether the concept described in the first two papers on CoWP is also valid for high-resolution LES simulations.

The Load center is introduced, for BEM and LES-AL, for estimating the position at which the total aggregated thrust force acts on the rotor plane. The calculation of the Load center is derived from the CoWP concept by replacing the wind velocity with the sectional thrust forces. For the LES-BL, this load position is given by the CoP. The Load center can be used to establish a connection between the flow-dependent CoWP and the turbine loads.

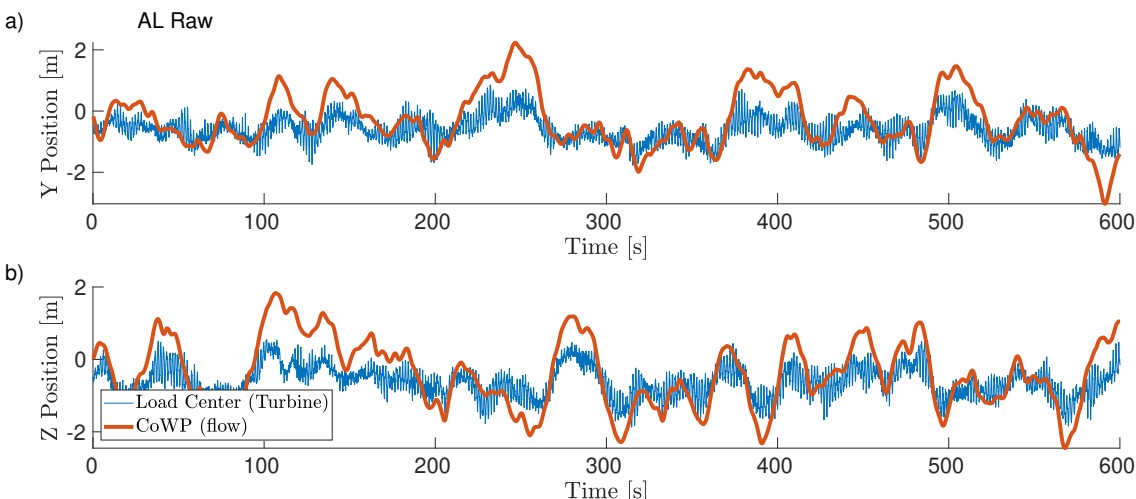

**Figure 19.** Time-series of the CoWP from LES in the rotor plane and the Load center of the LES-AL. Y component in a) and Z component in b).

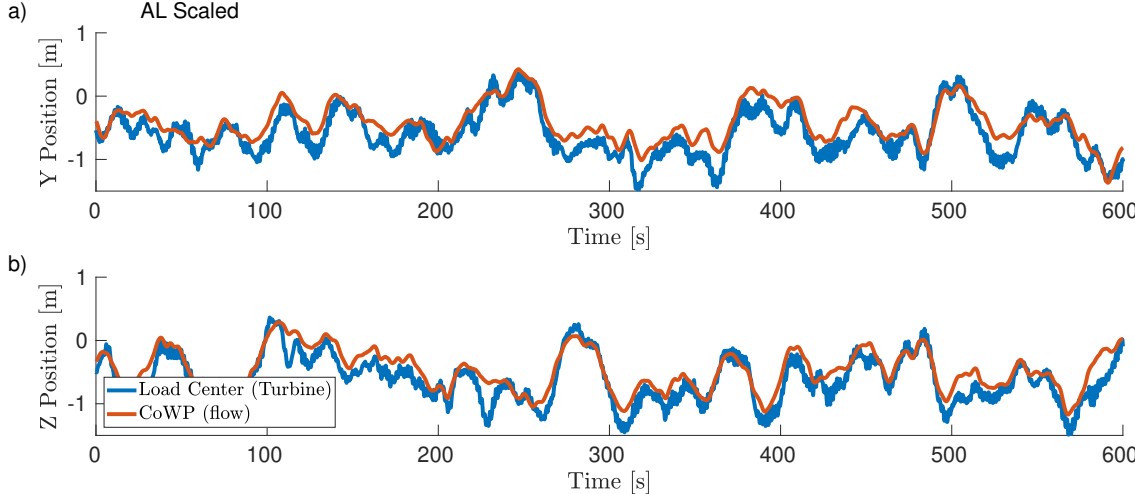

**Figure 20.** Time-series of the scaled CoWP from LES in the rotor plane and the filtered Load center of the LES-AL. Y component in a) and Z component in b). $r_{Pearson} = 0.908$.





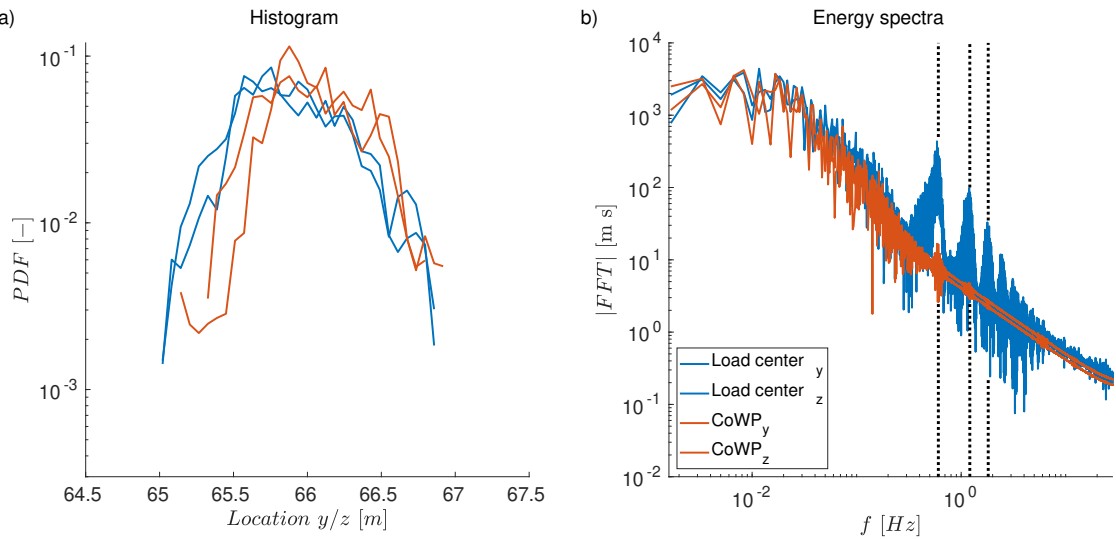

**Figure 21.** Statistical analysis (histogram in a) and energy spectra in b) of the scaled CoWP and the filtered Load center for the LES-AL simulation (figure 20). The dotted vertical lines represent the 3 P rotational frequency and higher harmonics of it.

From a laminar shear flow simulation, a turbine-specific calibration parameter can be determined. This single parameter summarises the relationship between the flow and the turbine loads. This methodology facilitates the prediction of load signals through the calculation of the CoWP in a turbulent wind field and subsequent scaling with the calibration parameter.

It has been shown that the methodology of using a calibration parameter derived from a laminar shear flow can also be applied to high-resolution LES simulations to scale the CoWP and obtain a load signal. In the LES-AL case, correlating the CoWP from the flow field just upstream of the turbine with the loads improves the agreement between the flow and the loads. This improvement arises because the interaction of the wind field with the induction zone is taken into account.

Nevertheless, further questions emerge directly from this work. Among them: What influence does the fluid-structure coupling of the blades have on the Load center and the CoWP? Does the CoWP analysis method work when two turbines are arranged in sequence, or when several turbines form a wind farm?

**Appendix A: Grid Study**

Several LES-AL simulations are carried out to determine the required grid resolution. The division of the refinement regions and the relative gradation to each other is kept the same. This makes it possible to vary the overall resolution through a single parameter in a comprehensible manner. The Power coefficient of the turbine ($C_P$) over the overall number of cells in shown in figure A1. From the 3.5 million cells mesh, the $C_P$ seems to be saturated. The same basic mesh is also used for the LES-BL simulations with a higher complexity. To be able to represent this complexity, the next finer mesh with 27.2 million cells was selected for the work.

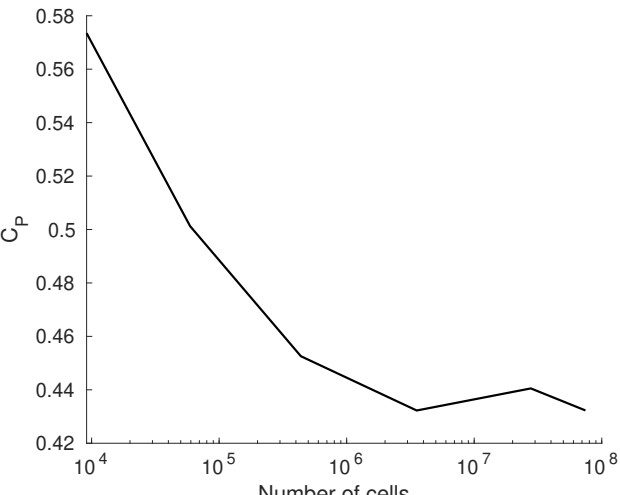

**Figure A1.** Power coefficient $C_P$ for different grid resolution in LES with LES-AL modelled turbine.

## Appendix B: Impact of the rotational direction in LES-AL

Figure B1 shows the time series of the Load center in a clockwise simulation (red) and a counterclockwise simulation (blue). At the start of the simulation, the velocity field still corresponds to the initial values everywhere. As the flow field around the rotor and the wake develops, the two simulations approach the final values within the first 40 s. This corresponds to an estimated wake size of roughly 2D, which corresponds to the near wake size (assuming the wake propagation speed is 55% of the freestream velocity at hub height). The Z component of the load centers then saturate for both directions of rotation to the value specified in section 4.2 (since the courses are identical, only one line is visible.). In the Y direction, saturation occurs in opposite directions, but with the same distance from the rotor center. As with the fluctuations in the laminar case (section 4.1), this shift in the Load center could be due to the smearing errors described in Churchfield et al. (2017). Since the focus of this work is on the analysis of the CoWP, this result is illustrated here as a property of the LES-AL method without further elaboration on the causes.

## Appendix C: Turbulent inflow method

In order to determine where the differences of the CoWP between the inflow and the LES (figure 15) are coming from, an analysis of the turbulent inflow method is done here. Therefore, a LES with a velocity jump is done. Figure C1 shows the time-averaged velocity field in the sectional view of such a simulation. Ten different velocity jumps in a range from -2 to 2 m/s are carried out. This range covers 99.6% of the fluctuations from the inflow field of the turbulent case from section 4.3.3 ($U_{mean}$ = 11.4 m/s, TI = 5%, Gaussian distribution of fluctuations).

Figure C2 a) shows the averaged velocity for the different velocity jumps. For a positive velocity jump, i.e. an acceleration,



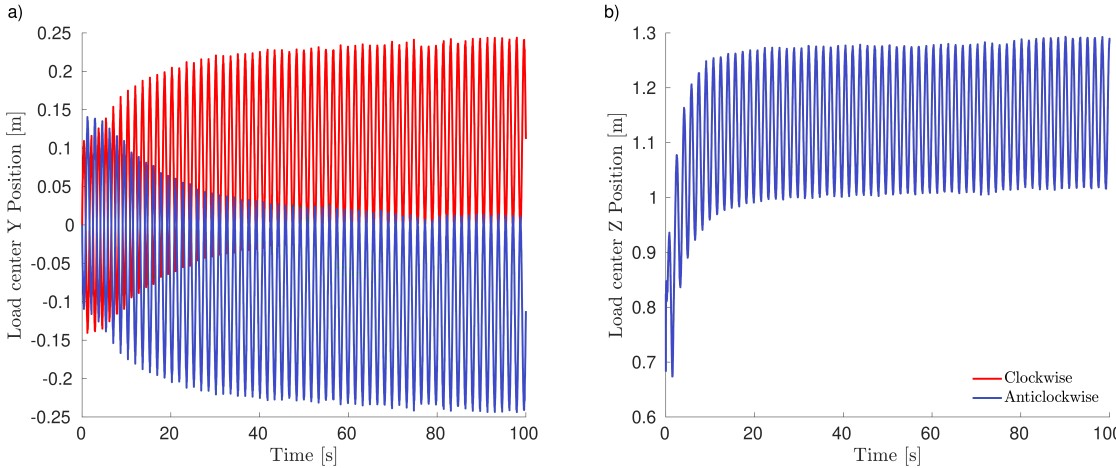

**Figure B1.** Time series of the Load center for LES-AL simulations with different rotational directions.

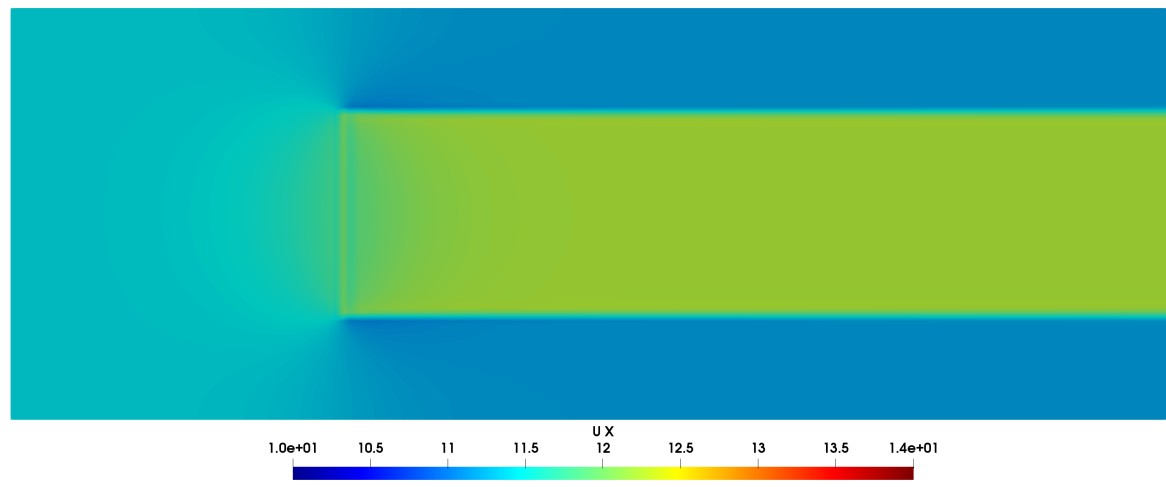

**Figure C1.** Time-averaged velocity field from a LES with velocity jump achieved by an actuator.





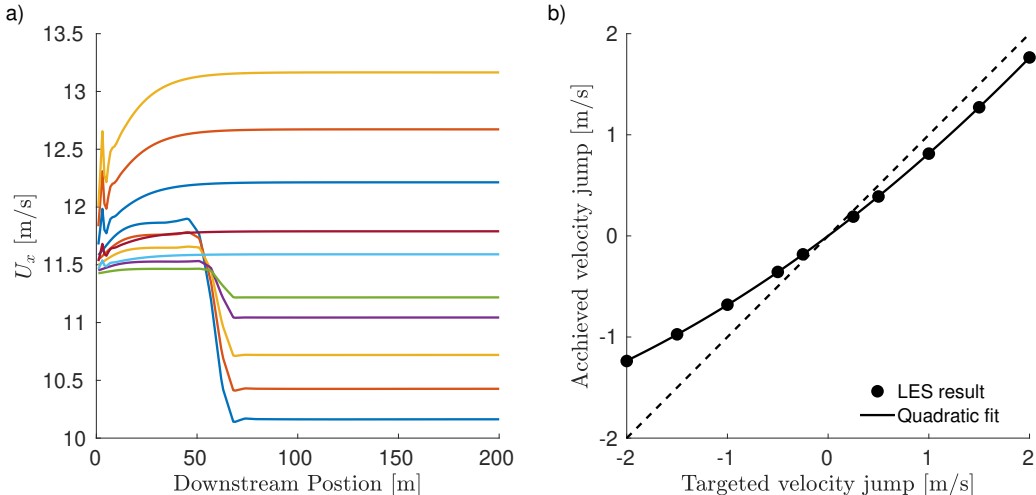

**Figure C2.** Time-averaged velocity over downstream position a) Scatter plot of the targeted velocity jump over achieved velocity jump b).

there is a kind of power law behaviour for reaching the target velocity. This is reached from 80 m after the inflow and remains constant until the outlet. With a negative velocity jump, i.e. a deceleration, the behaviour is different. Contrary to intuition, the velocity increases within the first 50 m and then drops sharply, but also reaches the target value after 80 m as the acceleration does.

Figure C2 b) shows a scatter plot of the targeted velocity jump over the achieved velocity jump. It can be seen that the absolute deviation for small jumps ($< 1$ m/s) is smaller than for larger ones. And that the achieved velocity jump diverges further from the target value for larger decelerations. The relationship between the achieved and targeted velocity jump can be represented by a quadratic fit, as shown with the black line.

Using the results from the simple simulations with velocity jumps, the CoWP curves of the input fields and the LES simulation are compared again qualitatively. Figure C3 a) and b) shows the CoWP curve for the input field in black and for the LES 225 metres after the inflow in green. Two points in time $t_1$ = 101 s and $t_2$ = 280 s are identified for further analysis. At both times, the $CoWP_Y$ is away from the center. At $t_1$, the total deviation between the LES and the input field is only 1.01 m and at $t_2$ 1.76 m.

Figures C3 c) and d) show velocity sections for the two times and the location of the CoWP for the inflow field. Figure C3 e) and f) show the same for the LES simulation. It is important to note that the TI has decreased during the transport through the domain (see figure 14). This can be easily recognised when comparing c) with e) by the fact that the range of velocities $> 12.5$ m/s in the LES is considerably smaller than in the input field. Furthermore, the range of velocities $< 11.4$ m/s at the right or right upper boundary is noticeably larger in the LES than in the input field. This can be explained by the fact that the Taylor hypothesis is only partially applicable (source). Due to transversal velocity components, transversal shifts occur.





**Figure C3.** Time series of the CoWP of the input wind field (black) and in LES after 225 m. Velocity plane of the wind field at t1 101s for the input field in c) and in LES in e). Velocity plane of the wind field at t2 280s for the input field in d) and in LES in f).





## Appendix D: Parameter study for the turbulent inflow

For a generalisation of the results of this work, simulations with different turbulence parameters are carried out here. Three different integral lengths (113 m, 126 m, 189 m) and three TI's (5%, 7.5%, 10%) are combined in BEM simulations. The procedure introduced in Section 4.3.1 is used for each combination. The time series of the Load center is filtered with a lowpass and the CoWP is scaled with the factor from table 3. Table D1 shows the values for the Pearson correlation coefficients.

**Table D1.** Correlation factor between Load center and CoWP for different turbulent fields in BEM simulations.

| L / TI | 5% | 7.5% | 10% |
|--------|-------|-------|-------|
| 113 m | 0.796 | 0.787 | 0.780 |
| 126 m | 0.814 | 0.807 | 0.804 |
| 189 m | 0.786 | 0.778 | 0.774 |

Due to the greater computational effort of LES-AL simulations, only one TI with three integral lengths are simulated. The Pearson correlation coefficients are presented in table D2.

**Table D2.** Correlation factor between Load center and CoWP for different turbulent fields in LES-AL simulations.

| L / TI | 5% |
|--------|-------|
| 113 m | 0.895 |
| 126 m | 0.908 |
| 189 m | 0.851 |

*Code and data availability.* All data created for this work was generated using open-source programmes. The data can be obtained from the authors upon request.

*Author contributions.* MB: Conceptualization, Methodology, simulations, data analysis and calculations, Writing – original draft. DM: Simulations, data analysis and calculations, writing – review and editing. JF: Review, analysis, discussion of the results, writing – review and editing. JP: Supervision, writing – review and editing.

*Competing interests.* An author (Joachim Peinke) is a member of the editorial board of Wind Energy Science journal.





*Acknowledgements.* This work was partially funded by the German Federal Ministry for Economic Affairs and Climate Action (BMWK) as part of the MOUSE project (FZK 03EE3067A) and the EU project FLOW (grant agreement N 101084205). Computational resources of the

University of Oldenburg were provided using the HPC cluster STORM, funded by the BMWK within the MOUSE project (FZK 03EE3067A). We acknowledge N. Manelil for constructive discussions and feedback in the early stages during the writing process.



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
