# Peer review of "Comparison of different simulation methods regarding loads, considering the Center of Wind Pressure"

_Wind Energy Science, 2025_

## Author Comment (AC1)

**Reply to referee comments**

Comparison of different simulation methods regarding loads, considering the Center of Wind Pressure

Referee's comments (RC) in blue

Autor's response (AC) in black

**Referee #1**

The paper is a continuation of the work by the group, around the concept of Center of Wind Pressure (CoWP). The interest of this concept was the fact that it is a turbine-independent quantity, it is only function of the wind velocity field, and it correlated with turbine loads. However, for it to be useful, two main questions needed answers:

a) How exactly does CoWP provide load information for a given turbine and

b) Is CoWP pertinent and applicable using different numerical approaches (BEM, LES-AL, DES-BL).

To address those questions, the authors used the (turbine dependent) Load Center as load-related quantity and compared with CoWP. The load center is a quantity that can be computed by all the methods employed (although table 2 suggests it cannot be done for DES-BL). They also used three different inflo cases: a laminar/uniform, a sheared and a turbulent flow. They verified that for the turbulent (and most useful and realistic) case, the simulations presented time-histories of CoWP that were nearly proportional to the Load Center. This turbine-dependent constant of proportionality was verified to be nearly the same for the laminar case, resulting in a way to simply compute it.

I believe the paper successfully answered those question, and opened others, for example the applicability in wind farms and in simulations involving fluid-structure interactions.

My only concern is about the form of presentation of the ideas. It was difficult to see, in a first glance, the point of the paper, by reading the abstract, for example. Also, in the introduction, the point was a little hidden and, only in the conclusion we could really appreciate the contribution. I would suggest a reformulation of those partes of the text.

As suggested by the referee we reformulated the Abstract and the Introduction to point out the scope of the paper in a clearer way.

Also, a few points should be clarified. For example, I would not call LES the methodology applied for the Blade-Resolved case, since, in essence, it was a DES-type of approach. Also, I I do not see why the Load Centre cannot be computed by DES-BL simulations (as indicated in table 2).

According to your comment we changed the label for the blade resolved results from LES-BL to DDES-BL.

For clarification of the Load center for BL section 3.5 was extended for a better explanation

I believe the scope of the paper and its results fit the purpose of the journal, but its form should be slightly improved before publishing.

**Referee #2**

The manuscript extends the new notion of "centre of wind pressure" (CoWP) which is a simple, but effective notion introduced previously by the same research group. In prior publications CoWP has only been tested against simple BEM simulations of wind turbines, whereas in the following work it is

assessed against simulations of various fidelities. This includes BEM, but also extends to LES in which the turbine is modelled through an actuator line method as well as a blade resolving DES. The paper reveals that indeed CoWP is suited to higher-fidelity simulations and a calibration factor is determined to link the flow structures within a turbulent inflow to the aerodynamic loads on the turbine, namely the moment applied to the shaft. Overall this manuscript is of a good scientific quality and ought to be published in Wind Energy Science where it will garner significant attention. I would suggest only minor modifications to the originally-submitted manuscript before publication.

**1. A lengthier discussion of the generation of the synthetic turbulence should be included. At some points in the manuscript it is a little opaque, for example on line 55 the spectrum is not defined, nor anywhere is the model spectrum that is used.**

As suggested by the referee we have added further information on the generation of the synthetic fields in Section 2.1. We have also defined the spectral tensor of the Mann model of inflow turbulence and provided an explicit form of the kinetic energy spectrum.

**2. A minor point but a 5MW, 126 diameter turbine does not really reflect the state-of-the-art in 2025.**

According to your comment we have changed the formulation in the text (line 201) to: "This model turbine is commonly used for scientific studies."

**3. In general the captions to the figures could be more informative. For example, figure 2 could mention the fact that the numbers reflect the spatial resolution in different regions of the domain. The legend of figure 3 is also obscured.**

We have checked all captions again and revised most of them (Figures 1, 2, 3, 7, 8, 11, 12, 14, 18, 19).

**4. The 3P frequency should be defined when it is introduced on line 288. The low-pass filter cut-off frequency should also be motivated. Why is a value of 110% of 3P chosen, for example?**

As suggested by the referee we added the definition of the 3P frequency and a description of the cut-off frequency.

Line 307 (previous 288): "Whereas the 3P frequency is defined as three times the rotational frequency of the turbine (=0.605Hz)."
Line 353: "This frequency was selected in order to filter out the high-frequency components while still capturing the dominant 3P frequency (plus an additional buffer of 10%)."

**5. Appendix C is sufficiently informative that it should be included in the main body of the text.**

We believe that including the entire Appendix C disrupts the storyline of the paper. Nevertheless, we understand the reviewer's approach. Therefore we have added a summary of the Appendix C to the main section at the appropriate place (section 4.3.2, line 387).

**6. There is a phantom (source) in line 464**

You are right we overlooked the missing source at this point. The necessary source has been added.

-> Jacobitz, F. G. and Schneider, K. (2024): Revisiting Taylor's hypothesis in homogeneous turbulent shear flow

---

## Author Response (AR2)

**Reply to referee comments**

Comparison of different simulation methods regarding loads, considering the Center of Wind Pressure

Referee's comments (RC) in blue

Autor's response (AC) in black

All minor (grammer) suggenstions where followed. We were just not shure what you ment by the citation formating in line 116. We corrected the name to "Kármán spectrum" and change the citation to the year. If this was not correct, we kindly ask you to have it checked by the style editors.